# ANALYTICAL MOMENT REGULARIZER FOR TRAINING ROBUST NETWORKS

## ABSTRACT

Despite the impressive performance of deep neural networks (DNNs) on numerous learning tasks, they still exhibit uncouth behaviours. One puzzling behaviour is the subtle sensitive reaction of DNNs to various noise attacks. Such a nuisance has strengthened the line of research around developing and training noise-robust networks. In this work, we propose a new training regularizer that aims to minimize the probabilistic expected training loss of a DNN subject to a generic Gaussian input. We provide an efficient and simple approach to approximate such a regularizer for arbitrarily deep networks. This is done by leveraging the analytic expression of the output mean of a shallow neural network, avoiding the need for memory and computation expensive data augmentation. We conduct extensive experiments on LeNet and AlexNet on various datasets including MNIST, CIFAR10, and CIFAR100 to demonstrate the effectiveness of our proposed regularizer. In particular, we show that networks that are trained with the proposed regularizer benefit from a boost in robustness against Gaussian noise to an equivalent amount of performing 3-21 folds of noisy data augmentation. Moreover, we empirically show on several architectures and datasets that improving robustness against Gaussian noise, by using the new regularizer, can improve the overall robustness against 6 other types of attacks by two orders of magnitude.

## 1 INTRODUCTION

Deep neural networks (DNNs) have emerged as generic models that can be trained to perform impressively well in a variety of learning tasks ranging from object recognition (He et al., 2016) and semantic segmentation (Long et al., 2015) to speech recognition (Hinton et al., 2012) and bioinformatics (Angermueller et al., 2016). Despite their increasing popularity, flexibility, generality, and performance, DNNs have been recently shown to be quite susceptible to small imperceptible input noise (Szegedy et al., 2014; Moosavi-Dezfooli et al., 2016; Goodfellow et al., 2015). Such analysis gives a clear indication that even state-of-the-art DNNs may lack robustness. Consequently, there has been an ever-growing interest in the machine learning community to study this uncanny behaviour. In particular, the work of (Goodfellow et al., 2015; Moosavi-Dezfooli et al., 2016) demonstrates that there are systematic approaches to constructing adversarial attacks that result in misclassification errors with high probability. Even more peculiarly, some noise perturbations seem to be doubly agnostic (Moosavi-Dezfooli et al., 2017), *i.e.* there exist deterministic perturbations that can result in misclassification errors with high probability when applied to different networks, irrespective of the input (denoted network and input agnostic).

Understanding this degradation in performance under adversarial attacks is of tremendous importance, especially for real-world DNN deployment, *e.g.* self-driving cars/drones and equipment for the visually impaired. A standard and popular means to alleviate this nuisance is noisy data augmentation in training, *i.e.* a DNN is exposed to noisy input images during training so as to bolster its robustness during inference. Several works have demonstrated that DNNs can in fact benefit from such augmentation (Moosavi-Dezfooli et al., 2016; Goodfellow et al., 2015). However, data augmentation in general might not be sufficient for two reasons. (1) Particularly with high-dimensional input noise, the amount of data augmentation necessary to sufficiently capture the noise space will be very large, which will increase training time. (2) Data augmentation with high energy noise can negatively impact the performance on noise-free test examples. This can be explained by the fundamental trade-off between accuracy and robustness (Tsipras et al., 2018; Boopathy et al., 2019). It can also arise from the fact that augmentation forces the DNN to have the same prediction for two vastly different versions of the same input, noise-free and a substantially corrupted version.

Therefore, in this paper, we propose a new regularizer for noise-robust networks to circumvent the aforementioned setbacks of data augmentation.

A natural objective for training against attacks sampled from a distribution $\mathcal{D}$, that bypasses the need for data augmentation, is the *expected loss* under this distribution. Since a closed-form expression is generally difficult to obtain or an approximate surrogate is expensive to evaluate (Monte Carlo estimates), we propose instead a closely related objective that is the loss of the *expected predictions* of the network under $\mathcal{D}$-distributed adversarial noise. Since it has been shown that Gaussian noise can be adversarial (Bibi et al., 2018) and that such noise is widely studied in applications such as image processing, we restrict the focus in this paper to the case where $\mathcal{D}$ is Gaussian. While this may seem to be too restrictive, we later show that improving the robustness of networks against Gaussian attacks also improves the robustness against a family of other types of attacks. However, even under such an assumption, only a memory and computationally expensive (expensive due to two-stage network linearization), closed-form approximate surrogate for network *expected predictions* exists (Bibi et al., 2018).

**Contributions.** **(i)** We formalize a new regularizer that is a function of the probabilistic first moment of the output of a DNN to train robust DNNs against noise sampled from distribution $\mathcal{D}$. **(ii)** Under the special choice of Gaussian attacks, *i.e.* $\mathcal{D}$ is Gaussian, we show how the first moment expression can be evaluated very efficiently during training for an arbitrary deep DNN by bypassing the need to perform memory and computationally expensive two-stage linearization. **(iii)** Extensive experiments using LeNet (LeCun et al., 1999) and AlexNet (Krizhevsky et al., 2012) architectures on MNIST (LeCun, 1998), CIFAR10, and CIFAR100 (Krizhevsky & Hinton, 2009) datasets demonstrate that a substantial enhancement in robustness can be achieved when using our regularizer in training. In fact, in the majority of the experiments, the improvement is better than training on the same dataset, augmented with 3 to 21 times Gaussian noisy data. Interestingly, the results suggest an excellent trade-off between accuracy and robustness. Moreover, we show that networks that are trained to be robust against Gaussian attacks using our proposed regularizer enjoy orders of magnitude boost in robustness against a family of other types of attacks.

## 2 RELATED WORK

Despite the impressive performance of DNNs on various tasks, they have been shown to be very sensitive to certain types of noise, commonly referred to as adversarial examples, particularly in the recognition task (Moosavi-Dezfooli et al., 2016; Goodfellow et al., 2015). Adversarial examples can be viewed as small imperceptible noise that, once added to the input of a DNN, its performance is severely degraded. This finding has incited interest in studying/measuring the robustness of DNNs.

The literature is rich with work that aims to unify and understand the notion of network robustness. For instance, Szegedy et al. (2014) suggested a spectral stability analysis for a wide class of DNNs by measuring the Lipschitz constant of the affine transformation describing a fully-connected or a convolutional layer. This result was extended to compute an upper bound for a composition of layers, *i.e.* a DNN. However, this measure sets an upper bound on the robustness over the *entire* input domain and does not take into account the noise distribution. Later, Fawzi et al. (2017a) defined robustness as the mean support of the minimum adversarial perturbation, which is now the most common definition for robustness. Not only was robustness studied against adversarial perturbations but also against geometric transformations to the input. Fawzi et al. (2018) emphasized the independence of the robustness measure to the ground truth class labels and that it should only depend on the classifier and the dataset distribution. Subsequently, two different metrics to measure DNN robustness were proposed: one for general adversarial attacks and another for noise sampled from uniform distribution. Recently, Gilmer et al. (2018) showed the trade-off between robustness and test error from a theoretical point of view on a simple classification problem with hyperspheres.

On the other hand, and based on various robustness analyses, several works proposed various approaches in building networks that are robust against noise sampled from well known distributions and against generic adversarial attacks. For instance, Grosse et al. (2017) proposed a model that was trained to classify adversarial examples with statistical hypothesis testing on the distribution of the dataset. Another approach is to perform statistical analysis on the latent feature space instead (Li & Li, 2017; Feinman et al., 2017), or train a DNN that rejects adversarial attacks (Lu et al., 2017). Moreover, the geometry of the decision boundaries of DNN classifiers was studied by Fawzi et al. (2017b) to infer a simple curvature test for this purpose. Using this method, one can restore the orig-

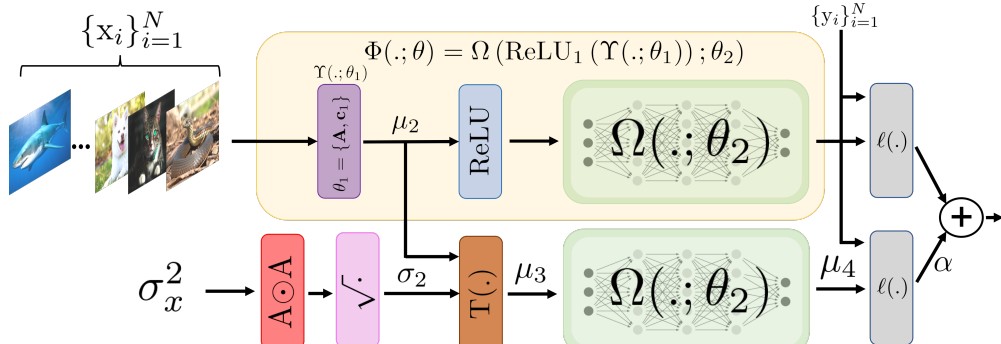

Figure 1: **Overview of the proposed graph for training Gaussian robust networks.** The yellow block corresponds to an arbitrary network $\Phi(.,\theta)$ viewed as the composition of two subnetworks separated by a ReLU. The stream on the bottom computes the output mean $\mu_4$ of the network $\Phi(:,\theta)$ assuming that (i) the noise input distribution is independent Gaussian with variances $\sigma_x^2$, and (ii) $\Omega(.\ :\ \theta_2)$ is approximated by a linear function. This evaluation for the output mean is efficient as it only requires an extra forward pass (bottom stream), as opposed to other methods that employ computationally and memory intensive network linearizations or data augmentation.

inal label and classify the input correctly. Restoring the original input using defense mechanisms, which can only detect adversarial examples, can be done by denoising (ridding it from its adversarial nature) so long as the noise perturbation is well-known and modeled apriori (Zhu et al., 2016). A fresh approach to robustness was proposed by Zantedeschi et al. (2017), where they showed that using bounded ReLUs (if augmented with Gaussian noise) to limit the output range can improve robustness. A different work proposed to distill the learned knowledge from a deep model to retrain a similar model architecture as a means to improving robustness (Papernot et al., 2016). This training approach is one of many adversarial training strategies for robustness Makhzani et al. (2016). More closely to our work is (Cisse et al., 2017), where a new training regularizer was proposed for a large family of DNNs. The proposed regularizer softly enforces that the upper bound of the Lipshitz constant of the output of the network to be less than or equal to one. Moreover and very recently, the work of Bibi et al. (2018) has derived analytic expressions for the output mean and covariance of networks in the form of (Affine, ReLU, Affine) under a generic Gaussian input. This work also demonstrates how a (memory and computation expensive) two-stage linearization can be employed to locally approximate a deep network with a two layer one, thus enabling the application of the derived expressions on the approximated shallower network.

Most prior work requires data augmentation, training new architectures that distill knowledge, or detect adversaries a priori, resulting in expensive training routines that may be ineffective in the presence of several input noise types. To this end, we address these limitations through our new regularizer that aims to fundamentally tackle Gaussian input noise without data augmentation and, as a consequence, improves overall robustness against other types of attacks.

## 3 METHODOLOGY

**Background on Network Moments.** Networks with a single hidden layer of the form (Affine, ReLU, Affine) can be written in the functional form $\mathbf{g}(\mathbf{x}) = \mathbf{B}\max(\mathbf{A}\mathbf{x} + \mathbf{c}_1, \mathbf{0}_p) + \mathbf{c}_2$. The $\max(.)$ is an element-wise operator, $\mathbf{A} \in \mathbb{R}^{p \times n}$, and $\mathbf{B} \in \mathbb{R}^{d \times p}$. Thus, $\mathbf{g} : \mathbb{R}^n \to \mathbb{R}^d$. Given that $\mathbf{x} \sim \mathcal{N}(\mu_x, \Sigma_x)$, Bibi et al. (2018) showed that:

**Theorem 1** *The first moment of $g(\mathbf{x})$ is* $\mathbb{E}[\mathbf{g}(\mathbf{x})] = \mathbf{B}\left(\mu_2 \odot \Phi\left(\frac{\mu_2}{\sigma_2}\right) + \sigma_2 \odot \varphi\left(\frac{\mu_2}{\sigma_2}\right)\right) + \mathbf{c}_2$.

Note that $\mu_2 = \mathbf{A}\mu_x + \mathbf{c}_1$, $\sigma_2 = \sqrt{\text{diag}(\Sigma_2)}$, $\Sigma_2 = \mathbf{A}\Sigma_x\mathbf{A}^\top$, $\Phi$ and $\varphi$ are the standard Gaussian cumulative (CDF) and density (PDF) functions, respectively. The vector multiplication and division are element-wise operations. Lastly, $\text{diag}(.)$ extracts the diagonal elements of a matrix into a vector. For ease of notation, we let $\mu_3 = T(\mu_2, \sigma_2) = (\mu_2 \odot \Phi(\frac{\mu_2}{\sigma_2}) + \sigma_2 \odot \varphi(\frac{\mu_2}{\sigma_2}))$.

To extend the results of Theorem (1) to deeper models, a two-stage linearization was proposed in Bibi et al. (2018), where $(\mathbf{A}, \mathbf{B})$ and $(\mathbf{c}_1, \mathbf{c}_2)$ are taken to be the Jacobians and biases of the first

order Taylor approximation to the two network functions around a ReLU layer in a DNN. Refer to Bibi et al. (2018) for more details about this expression and the proposed linearization.

**Proposed Robust Training Regularizer.** To propose an alternative to noisy data augmentation to address its drawbacks, one has to realize that this augmentation strategy aims to minimize the expected training loss of a DNN when subjected to noisy input distribution $\mathcal{D}$ through sampling. In fact, it minimizes an empirical loss that approximates this expected loss when enough samples are present during training. When sampling is insufficient (a drawback of data augmentation in high-dimensions), this approximation is too loose and robustness can suffer. However, if we have access to an analytic expression for the expected loss, expensive data augmentation can be averted. This is the key motivation of the paper. Mathematically, the training loss can be modeled as

$$\min_{\theta} \ \sum_{i=1}^{N} \Big( \ell \left( \Phi(\mathbf{x}_i; \theta), y_i \right) + \alpha \, \mathbb{E}_{\mathbf{n} \sim \mathcal{D}} \left[ \ell \left( \Phi(\mathbf{x}_i + \mathbf{n}; \theta), y_i \right) \right] \Big). \tag{1}$$

Here, $\Phi : \mathbb{R}^n \to \mathbb{R}^d$ is any arbitrary network with parameters $\theta$, $\ell$ is the loss function, $\{(\mathbf{x}_i, y_i)\}_{i=1}^{N}$ are the noise-free data-label training pairs, and $\alpha \geq 0$ is a trade off parameter. While the first term in Equation 1 is the standard empirical loss commonly used for training, the second term is often replaced with its Monte Carlo estimate through data augmentation. That is, for each training example $\mathbf{x}_i$, the second term is approximated with an empirical average of $\tilde{N}$ noisy examples of $\mathbf{x}_i$ such that $\mathbb{E}_{\mathbf{n} \sim \mathcal{D}}[\ell \left( \Phi(\mathbf{x}_i + \mathbf{n}; \theta), y_i \right)] \approx \frac{1}{\tilde{N}} \sum_{j=1}^{\tilde{N}} \ell \left( \Phi(\mathbf{x}_i + \mathbf{n}_j; \theta), y_i \right)$. This will increase the size of the dataset by a factor of $\tilde{N}$, which will in turn increase training complexity. As discussed earlier, network performance on the noise-free examples can also be negatively impacted. Note that obtaining a closed form expression for the second term in Equation 1 for some of the popularly used losses $\ell$ is more complicated than deriving expressions for the output mean of the network $\Phi$ itself, *e.g.* in Theorem (1). Therefore, we propose to replace this loss with the following surrogate

$$\min_{\theta} \ \sum_{i=1}^{N} \Big( \ell \left( \Phi(\mathbf{x}_i; \theta), y_i \right) + \alpha \ell \left( \mathbb{E}_{\mathbf{n} \sim \mathcal{D}} \left[ \Phi(\mathbf{x}_i + \mathbf{n}; \theta) \right], y_i \right). \Big) \tag{2}$$

Because of Jensen's inequality, Equation 2 is a lower bound to Eq Equation 1 when $\ell$ is convex, which is the case for most popular losses including $\ell_2$-loss and cross-entropy loss. The proposed second term in Equation 2 encourages that the output mean of the network $\Phi$ of every noisy example $(\mathbf{x}_i + \mathbf{n})$ matches the correct class label $y_i$. This regularizer will stimulate a separation among the output mean of the classes if the training data is subjected to noise sampled from $\mathcal{D}$. Having access to an analytic expression for these means will prompt a simple inexpensive training, where the actual size of the training set is unaffected and augmentation is avoided. This form of regularization is proposed to replace data augmentation.

While a closed-form expression for the second term of Equation 2 might be infeasible for a general network $\Phi(.)$, an expensive approximation can be attained. In particular, Theorem Equation 1 provides an analytic expression to evaluate the second term in Equation 2, for when $\mathcal{D}$ is Gaussian and when the network is approximated by a two-stage linearization procedure as $\Phi(\mathbf{x}) \approx \mathbf{B} \max \left( \mathbf{A}\mathbf{x} + \mathbf{c}_1, \mathbf{0}_p \right) + \mathbf{c}_2$. However, it is not clear how to utilize such a result to regularize networks during training with Equation 2 as a loss. This is primarily due to the computationally expensive and memory intensive network linearization proposed in Bibi et al. (2018). Specifically, the linearization parameters $(\mathbf{A}, \mathbf{B}, \mathbf{c}_1, \mathbf{c}_2)$ are a function of the network parameters, $\theta$, which are updated with every gradient descent step on Equation 2; thus, two-stage linearization has to be performed in every $\theta$ update step, which is infeasible.

**On an Efficient Approximation to Equation 2.** The loss in Equation 2 proposes a generic approach to train robust arbitrary networks against noise sampled from an arbitrary distribution $\mathcal{D}$. Since the problem in its general setting is too broad for detailed analysis, we restrict the scope of this work to the class of networks, which are most popularly used and parameterized by $\theta$, $\Phi(.; \theta) : \mathbb{R}^n \to \mathbb{R}^d$ with ReLUs as nonlinear activations. Moreover, since random Gaussian noise was shown to exhibit an adversarial nature Bibi et al. (2018); Rauber et al. (2017); Franceschi et al. (2018), and it is one of the most well studied noise models for the useful properties it exhibits, we restrict $\mathcal{D}$ to the case of Gaussian noise. In particular, $\mathcal{D}$ is independent zero-mean Gaussian noise at the input, *i.e.* $\mathbf{n} \sim \mathcal{D} = \mathcal{N} \left( \mathbf{0}, \Sigma_x = \text{Diag} \left( \sigma_x^2 \right) \right)$, where $\sigma_x^2 \in \mathbb{R}^n$ is a vector of variances and Diag(.) reshapes the vector elements into a diagonal matrix. Generally, it is still difficult to compute the second

term in Equation 2 under Gaussian noise for arbitrary networks. However, if we have access to an inexpensive approximation of the network, avoiding the computationally and memory expensive network linearization in Bibi et al. (2018), an approximation to the second term in Equation 2 can be used for efficient robust training directly on $\theta$.

Consider the $l^{\text{th}}$ ReLU layer in $\Phi(.;\theta)$. the network can be expressed as $\Phi(.;\theta) = \Omega(\text{ReLU}_l(\Upsilon(.,\theta_1));\theta_2)$. Note that the parameters of the overall network $\Phi(.;\theta)$ is the union of the parameters of the two subnetworks $\Upsilon(.;\theta_1)$ and $\Omega(.;\theta_2)$, $i.e.$ $\theta = \theta_1 \cup \theta_2$. Throughout this work and to simplify the analysis, we set $l = 1$. With such a choice of $l$, the first subnetwork $\Upsilon(.,\theta_1)$ is affine with $\theta_1 = \{\mathbf{A}, \mathbf{c}_1\}$. However, the second subnetwork $\Omega(.,\theta_2)$ is not linear in general, and thus, one can linearize $\Omega(.,\theta_2)$ at $\mathbb{E}_{\mathbf{n}\sim\mathcal{N}(\mathbf{0},\Sigma_x)}[\text{ReLU}_1(\Upsilon(\mathbf{x}_i + \mathbf{n};\theta_1))] = T(\mu_2,\sigma_2) = \mu_3$. Note that $\mu_3$ is the output mean after the ReLU and $\mu_2 = \mathbf{A}\mathbf{x}_i + \mathbf{c}_1$, since $\Upsilon(\mathbf{x}_i + \mathbf{n};\theta_1) = \mathbf{A}(\mathbf{x}_i + \mathbf{n}) + \mathbf{c}_1$. Both $T(.,.)$ and $\sigma_2$ are defined in Equation 1. Thus, linearizing $\Omega$ at $\mu_3$ with linearization parameters $(\mathbf{B}, \mathbf{c}_2)$ being the Jacobian of $\Omega$ and $\mathbf{c}_2 = \Omega(\mu_3,\theta_2) - \mathbf{B}\mu_3$, we have that, for any point $\mathbf{v}_i$ close to $\mu_3$: $\Omega(\mathbf{v}_i,\theta_2) \approx \mathbf{B}\mathbf{v}_i + \mathbf{c}_2$. While computing $(\mathbf{B}, \mathbf{c}_2)$ through linearization is generally very expensive, computing the approximation to Equation 2 requires $explicit$ access to neither $\mathbf{B}$ nor $\mathbf{c}_2$. Note that this second term for $l = 1$ is given as:

$$\ell\left(\mathbb{E}_{\mathbf{n}\sim\mathcal{N}(\mathbf{0},\Sigma_x)}[\Phi(\mathbf{x}_i + \mathbf{n};\theta)], y_i\right) = \ell\left(\mathbb{E}_{\mathbf{z}_i\sim\mathcal{N}(\mathbf{x}_i,\Sigma_x)}[\Omega\left(\text{ReLU}_1\left(\Upsilon(\mathbf{z}_i;\theta_1)\right);\theta_2\right)], y_i\right)$$
$$= \ell\left(\mathbb{E}_{\mathbf{z}_i\sim\mathcal{N}(\mathbf{x}_i,\Sigma_x)}[\Omega\left(\text{ReLU}_1\left(\mathbf{A}\mathbf{z}_i + \mathbf{c}_1\right);\theta_2\right)], y_i\right)$$
$$\approx \ell\left(\mathbb{E}_{\mathbf{z}_i\sim\mathcal{N}(\mathbf{x}_i,\Sigma_x)}\left[\mathbf{B}\left(\text{ReLU}_1\left(\mathbf{A}\mathbf{z}_i + \mathbf{c}_1\right)\right) + \mathbf{c}_2\right], y_i\right) \tag{3}$$
$$= \ell\left(\mathbf{B}\mu_3 + \mathbf{c}_2, y_i\right) = \ell\left(\Omega(\mu_3,\theta_2), y_i\right).$$

The approximation follows from the assumption that the input to the second subnetwork $\Omega(.;\theta_2)$, $i.e.$ $\mathbf{v}_i = \text{ReLU}_1(\mathbf{A}\mathbf{z}_i + \mathbf{c}_1))$, is close to the point of linearization $\mu_3$ such that $\Omega(\mathbf{v}_i;\theta_2) \approx \mathbf{B}\mathbf{v}_i + \mathbf{c}_2$. Or simply, that the input to $\Omega$ is close to the mean inputs, $i.e.$ $\mu_3$, to $\Omega$ under Gaussian noise. The penultimate equality follows from the linearity of the expectation. As for the last equality, $(\mathbf{B}, \mathbf{c}_2)$ are the linearization parameters of $\Omega$ at $\mu_3$, where $\mathbf{c}_2 = \Omega(\mu_3,\theta_2) - \mathbf{B}\mu_3$ by the first order Taylor approximation. Thus, computing the second term of Equation 2 according to Equation 3 can be simply approximated by a forward pass of $\mu_3$ through the second network $\Omega$. As for computing $\mu_3 = T(\mu_2,\sigma_2)$, note that $\mu_2 = \mathbf{A}\mathbf{x}_i + \mathbf{c}_1$ in Equation 3, which is equivalent to a forward pass of $\mathbf{x}_i$ through the first subnetwork because $\Upsilon(.,\theta_1)$ is linear with $\theta_1 = \{\mathbf{A}, \mathbf{c}_1\}$. Moreover, since $\sigma_2 = \sqrt{\text{diag}\left(\mathbf{A}\Sigma_x\mathbf{A}^\top\right)}$, we have: $\sigma_2 = \sqrt{\text{diag}\left(\mathbf{A}\text{Diag}\left(\sigma_x^2\right)\mathbf{A}^\top\right)} = \sqrt{(\mathbf{A}\odot\mathbf{A})\sigma_x^2}$. The expression for $\sigma_2$ can be efficiently computed by simply squaring the linear parameters in the first subnetwork and performing a forward pass of the input noise variance $\sigma_x^2$ through $\Upsilon$ without the bias $\mathbf{c}_1$ and taking the element-wise square root. Lastly, it is straightforward to compute $T(\mu_2,\sigma_2)$ as it is an element-wise function in Equation 1. The overall computational graph in Figure 1 shows a summary of the computation needed to evaluate the loss in Equation 2 using only forward passes through the two subnetworks $\Upsilon$ and $\Omega$. It is now possible with the proposed efficient approximation of our proposed regularizer in Equation 2 to efficiently train networks on noisy training examples that are corrupted with noise $\mathcal{N}(\mathbf{0},\Sigma_x)$ without any form of prohibitive data augmentation.

## 4 EXPERIMENTS

In this section, we conduct experiments on multiple network architectures and datasets to demonstrate the effectiveness of our proposed regularizer in training more robust networks, especially in comparison with data augmentation. To standardize robustness evaluation, we first propose a new unified robustness metric against additive noise from a general distribution $\mathcal{D}$ and later specialize it when $\mathcal{D}$ is Gaussian. Lastly, we show that networks trained with our proposed regularizer not only outperform in robustness networks trained with Gaussian augmented data. Moreover, we show that such networks are also much more magnitudes times robust against other types of attacks.

**On the Robustness Evaluation Metric.** While there is a consensus on the definition of robustness in the presence of adversarial attacks, as the smallest perturbation required to $fool$ a network, $i.e.$ to change its prediction, it is not straightforward to extend such a definition to additive noise sampled from a distribution $\mathcal{D}$. In particular, the work of Fawzi et al. (2018) tried to address this difficulty by defining the robustness of a classifier around an example $\mathbf{x}$ as the distance between $\mathbf{x}$ and the closest

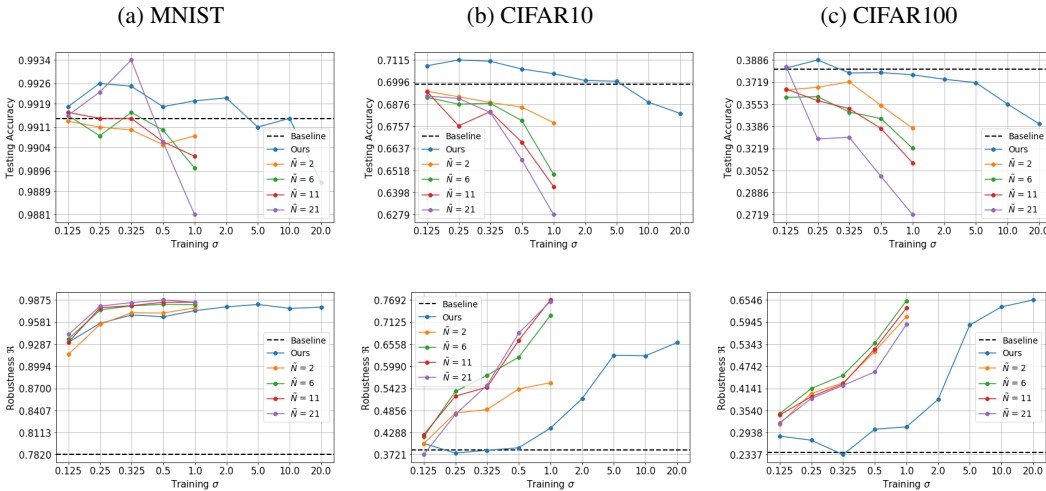

Figure 2: **General trade-off between accuracy and robustness on LeNet**. We see, in all plots, that the accuracy tends to be negatively correlated with robustness over varying noise levels and amount of augmentation. Baseline refers to training with neither data augmentation nor our regularizer. However, it is hard to compare the performance of our method against data augmentation from these plots as we can only compare the robustness of models with similar noise-free testing accuracy.

decision boundary. However, this definition is difficult to compute in practice and is not scalable, as it requires solving a generally nonconvex optimization problem for every testing example $\mathbf{x}$ that may also suffer from poor local minima. To remedy these drawbacks, we present a new robustness metric for generic additive noise.

**Robustness Against Additive Noise.** Consider a classifier $\Psi(.)$ with $\psi(\mathbf{x}) = \arg\max_i \Psi_i(\mathbf{x})$ as the predicted class label for the example $\mathbf{x}$ regardless of the correct class label $y_i$. We define the robustness on a sample $\mathbf{x}$ against a generic additive noise sampled from a distribution $\mathcal{D}$ as

$$\Re_{\mathcal{D}}(\mathbf{x}) = \mathbb{P}_{\mathbf{n} \sim \mathcal{D}}\{\psi(\mathbf{x} + \mathbf{n}) = \psi(\mathbf{x})\}. \tag{4}$$

Here, the proposed robustness metric $\Re_{\mathcal{D}}(\mathbf{x})$ measures the probability of the classifier to preserve the original prediction of the noise-free example $\psi(\mathbf{x})$ after adding noise, $\psi(\mathbf{x} + \mathbf{n})$, from distribution $\mathcal{D}$. Therefore, the robustness over a testing dataset $\mathcal{T}$ can be defined as the expected robustness over the test dataset: $\Re_{\mathcal{D}}(\mathcal{T}) = \mathbb{E}_{\mathbf{x} \sim \mathcal{T}}[\Re_{\mathcal{D}}(\mathbf{x})]$. Inspired by Franceschi et al. (2018), for ease, we relax Equation 4 from the probability of preserving the prediction score to a 0/1 robustness over $m$-randomly sampled examples from $\mathcal{D}$. That is, $\Re_{\mathcal{D}}(\mathbf{x}) = 1$ means that, among $m$ randomly sampled noise from $\mathcal{D}$ added to $\mathbf{x}$, none changed the prediction from $\psi(\mathbf{x})$. However, if a single example of these $m$ samples changed the prediction from $\psi(\mathbf{x})$, we set $\Re_{\mathcal{D}}(\mathbf{x}) = 0$. Thus, the robustness score is the average of this measure over the testing dataset $\mathcal{T}$.

**Robustness Against Gaussian Noise.** For additive Gaussian noise, *i.e.* $\mathcal{D} = \mathcal{N}(\mathbf{0}, \Sigma_x = \text{Diag}(\sigma_x^2))$, robustness is averaged over a range of testing variances $\sigma_x^2$. We restrict $\sigma_x$ to 30 evenly sampled values in $[0, 0.5]$, where this set is denoted as $\mathcal{A}$[1]. In practice, this is equivalent to sampling $m$ Gaussian examples for each $\sigma_x \in \mathcal{A}$, and if none of the $m$ samples changes the prediction of the classifier $\psi$ from the original noise-free example, the robustness for that sample at that $\sigma_x$ noise level is set to 1 and then averaged over the complete testing set. Then, the robustness is the average over multiple $\sigma_x \in \mathcal{A}$. To make the computation even more efficient, instead of sampling a large number of Gaussian noise samples $(m)$, we only sample a single noise sample with the average energy over $\mathcal{D}$. That is, we sample a single $\mathbf{n}$ of norm $\|\mathbf{n}\|_2 = \sigma_x \sqrt{n}$. This is due to the fact that $\mathbb{E}_{(\mathbf{0}_n, \sigma_x^2 \mathbf{I})}[\|\mathbf{x}\|_2] = \sqrt{2}\sigma_x \Gamma\left(\frac{n+1}{2}\right)/\Gamma\left(\frac{n}{2}\right) \overset{n \to \infty}{=} \sigma_x \sqrt{n}$.

**Experimental Setup.** In this section, we demonstrate the effectiveness of the proposed regularizer in improving robustness. Several experiments are performed with our objective Equation 2, where we strike a comparison with data augmentation approaches.

**Architecture Details.** The input images in MNIST (gray-scale) and CIFAR (colored) are squares with sides equal to 28 and 32, respectively. Since AlexNet was originally trained on ImageNet of

---

[1]We assume that the input $\mathbf{x}$ is normalized $[0, 1]^n$

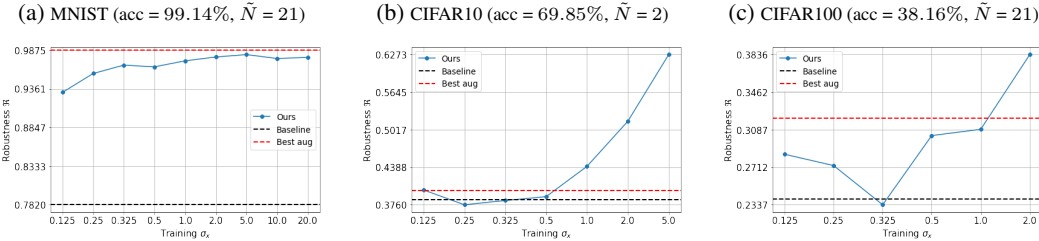

Figure 3: **Fair robustness comparison of LeNet with data augmentation and our regularizer**. We only report results for models with a test accuracy that is at least as good as the accuracy of the baseline with a tolerance: $0\%$, $0.39\%$, and $0.75\%$ for MNIST, CIFAR10, CIFAR100, respectively. Only the models with the highest robustness are presented. Training with our regularizer can attain similar/better robustness than 21-fold noisy data augmentation on MNIST and CIFAR100, while maintaining a high noise-free test accuracy.

sides equal to 224, we will marginally alter the implementation of AlexNet in TorchVision Marcel & Rodriguez (2010) to accommodate for this difference. First, we change the number of hidden units in the first fully-connected layer (in LeNet to $4096$, AlexNet to $256$, LeNet on MNIST to $3136$). For AlexNet, we changed all pooling kernel sizes from 3 to 2 and the padding size of conv1 from 2 to 5. Second, we swapped each maxpool with the preceding ReLU, which makes training and inference more efficient. Third, we enforce that the first layer in all the models is a convolution followed by ReLU as discussed earlier. Lastly, to simplify analysis, we removed all dropout layers. We leave the details of the optimization hyper-parameters to the **appendix**.

**Results.** For each model and dataset, we compare baseline models, *i.e.* models trained with noise-free data and without our regularization, with two others: one using data augmentation and another using our proposed regularizer. Each of the latter has two configurable variables: the level of noise controlled by $\sigma_x^2$ during training, and the amount of noise controlled by the trade-off coefficient $\alpha$ in Equation 2 or $\tilde{N}$ (number of added noisy training examples) in the case of augmentation.

**Accuracy vs. Robustness.** We start by demonstrating that data augmentation tends to improve the robustness, as captured by $\Re(\mathcal{T})$ over the test set, at the expense of decreasing the testing accuracy on the noise-free examples. Realizing this is essential for a fair comparison, as one would need to compare the robustness of networks that only have similar noise-free testing accuracies. To this end, we ran 60 training experiments with data augmentation on LeNet with three datasets (MNIST, CIFAR10, and CIFAR100), four augmentation levels ($\tilde{N} \in \{2, 6, 11, 21\}$), and five noise levels ($\sigma_x \in \mathcal{A} = \{0.125, 0.25, 0.325, 0.5, 1.0\}$). In contrast, we ran robust training experiments using Equation 2 with the trade-off coefficient $\alpha \in \{0.5, 1, 1.5, 2, 5, 10, 20\}$ on the same datasets, but we extended the noise levels $\sigma_x$ to include the extreme noise regime of $\sigma_x \in \{2, 5, 10, 20\}$. These noise levels are too large to be used for data augmentation, especially since $\mathbf{x} \in [0, 1]^n$; however, as we will see, they are still beneficial for our proposed regularizer. Figure 2 shows both the testing accuracy and robustness as measured by $\Re(\mathcal{T})$ over a varying range of training $\sigma_x$ for the data augmentation approach of LeNet on MNIST, CIFAR-10 and CIFAR-100. It is important to note here that the main goal of these plots is not to compare the robustness score, but rather, to demonstrate a very important trend. In particular, increasing the training $\sigma_x$ for each approach degrades testing accuracy on noise-free data. However, the degradation in our approach is much more graceful since the trained LeNet model was never directly exposed to individually corrupted examples during training as opposed to the data augmentation approach. Note that our regularizer enforces the separation between the expected output prediction analytically. Moreover, the robustness of both methods consistently improves as the training $\sigma_x$ increases. This trend holds even on the easiest dataset (MNIST). Interestingly, models trained with our regularizer enjoy an improvement in testing accuracy over the baseline model. Such behaviour only emerges with a large factor of augmentation, $\tilde{N} = 21$, and a small enough training $\sigma_x$ on MNIST. This indicates that models can benefit from better accuracy with a good approximation of Equation 1 through our proposed objective or through extensive Monte Carlo estimation. However, as $\sigma_x$ increases, Monte Carlo estimates of the second term in Equation 1 via data augmentation (with $\tilde{N} = 21$) is no longer enough to capture the noise.

**Robustness Comparison.** For fair comparison, it is essential to only compare the robustness of networks that achieve similar testing accuracy, since perfect robustness is attainable with a deterministic classifier that assigns the same class label regardless of the input. In fact, we proposed a unified robustness metric for the reason that most commonly used metrics are disassociated from

| | $\sigma$ | PGD | LBFGS | FGSM | DF2 | GNR | ACC |
|---|---|---|---|---|---|---|---|
| LeNet on MNIST | 0 | $1.02 \times 10^{-03}$ | $5.50 \times 10^{-04}$ | $2.24 \times 10^{-02}$ | $5.91 \times 10^{-04}$ | 77.45 | 98.50 |
| | 0.125 | $3.36 \times 10^{-01}$ | $3.43 \times 10^{-01}$ | $8.19 \times 10^{-01}$ | $2.05 \times 10^{-01}$ | 93.14 | 97.50 |
| | 0.250 | $4.58 \times 10^{-01}$ | $4.31 \times 10^{-01}$ | 1.21 | $2.63 \times 10^{-01}$ | 95.64 | 98.75 |
| | 0.325 | $4.21 \times 10^{-01}$ | $4.51 \times 10^{-01}$ | 1.17 | $2.33 \times 10^{-01}$ | 96.75 | 97.50 |
| | 1.0 | $5.44 \times 10^{-01}$ | $5.22 \times 10^{-01}$ | 1.34 | $2.95 \times 10^{-01}$ | 97.32 | 99.00 |
| AlexNet on CIFAR100 | 0 | $2.30 \times 10^{-05}$ | $2.50 \times 10^{-05}$ | $1.50 \times 10^{-05}$ | $2.10 \times 10^{-05}$ | 29.69 | 34.75 |
| | 0.12 | $3.64 \times 10^{-04}$ | $2.83 \times 10^{-04}$ | $5.06 \times 10^{-04}$ | $2.16 \times 10^{-04}$ | 31.65 | 33.50 |
| | 0.250 | $4.37 \times 10^{-04}$ | $3.86 \times 10^{-04}$ | $6.50 \times 10^{-04}$ | $2.47 \times 10^{-04}$ | 32.85 | 32.25 |
| | 0.325 | $5.37 \times 10^{-04}$ | $4.04 \times 10^{-04}$ | $7.29 \times 10^{-04}$ | $3.18 \times 10^{-04}$ | 33.84 | 34.25 |
| | 1.0 | $4.92 \times 10^{-04}$ | $3.26 \times 10^{-04}$ | $6.50 \times 10^{-04}$ | $2.85 \times 10^{-04}$ | 34.65 | 35.50 |

Table 1: **Gaussian robustness improves overall robustness.** We report the robustness metrics corresponding to various attacks (PGD, LBFGS, FGSM, and DF2), our proposed GNR metric, and the test accuracy ACC for LeNet and AlexNet networks trained on MNIST and CIFAR100 using our proposed regularizer with noise variance $\sigma$ in training. Note that $\sigma = 0$ corresponds to baseline models trained without our regularizer. We observe that training networks with our proposed regularizer (designed for additive Gaussian attacks) not only improves the robustness against Gaussian attacks but also against 6 other types of attacks which 4 of them listed here and the others are left for **appendix**.

the ground-truth labels and only consider model predictions. Therefore, we filtered out the results from Figure 2 by removing all the experiments that achieved lower test accuracy than the baseline model. Figure 3 summarizes these results for LeNet. Now, we can clearly see the difference between training with data augmentation and our approach. For MNIST (Figure 3a), we achieved the same robustness as 21-fold data augmentation without feeding the network with any noisy examples during training and while preserving the same baseline accuracy. Interestingly, for CIFAR10 (Figure 3b), our method is twice as robust as the best robustness achieved via data augmentation. Moreover, for CIFAR100 (Figure 3c), we are able to outperform data augmentation by around 5%. Finally, for extra validation, we also conducted the same experiments with AlexNet on CIFAR10 and CIFAR100 which can be found in the **appendix**. We can see that our proposed regularizer can improve robustness by 15% on CIFAR10 and around 25% on CIFAR100. It is interesting to note that for CIFAR10, data augmentation could not improve the robustness of the trained models without drastically degrading the testing accuracy on the noise-free examples. Moreover, it is interesting to observe that the best robustness achieved through data augmentation is even worse than the baseline. This could be due to the trade-off coefficient $\alpha$ in Equation 1.

**Towards General Robustness via Gaussian Robustness.** Here, we investigate whether improving robustness to Gaussian input noise can improve robustness against other types of attacks. Specifically, we compare the robustness of models trained using our proposed regularizer (robust again Gaussian attacks) with baseline models subject to different types of attacks: Projected Gradient Descent (PGD) and LBFGS attacks Szegedy et al. (2014), Fast Sign Gradient Method (FGSM) Goodfellow et al. (2015), and DeepFool L2Attack (DF2) Moosavi-Dezfooli et al. (2016) as provide by Rauber et al. (2017). For all these attacks, we report the minimum energy perturbation that can change the network prediction. We also report our Gaussian Network Robustness (GNR) metric, which is the Gaussian version of Equation 4 along with the testing accuracy (ACC). We perform experiments on LeNet on MNIST, CIFAR10 and CIFAR100 datasets and on AlexNet on both CIFAR10 and CIFAR100. Due to space constraints, we show the robustness results for only LeNet on MNIST and AlexNet of CIFAR100 and leave the rest along with two other types of attacks for the **appendix**. Table 1 shows that improving GNR through our data augmentation free regularizer can significantly improve all robustness metrics. For instance, comparing LeNet trained with our proposed regularizer against LeNet trained without any regularization, *i.e.* $\sigma = 0$, we see that robustness against all types of attacks improves by almost two orders of magnitude, while maintaining a similar testing accuracy. A similar improvement in performance is consistently present for AlexNet on CIFAR100.

## 5 CONCLUSION

Addressing the sensitivity problem of deep neural networks to adversarial perturbation is of great importance to the machine learning community. However, building robust classifiers against this noises is computationally expensive, as it is generally done through the means of data augmentation. We propose a generic lightweight analytic regularizer, which can be applied to any deep neural network with a ReLU activation after the first affine layer. It is designed to increase the robustness of the trained models under additive Gaussian noise. We demonstrate this with multiple architectures and datasets and show that it outperforms data augmentation without observing any noisy examples.

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

## A    EXPERIMENTAL SETUP AND DETAILS.

All experiments, are conducted using PyTorch version 0.4.1 Paszke et al. (2017). All hyper-parameters are fixed and Table 2 we report the setup for the two optimizers. In particular, we use the Adam optimizaer Kingma & Ba (2015) with $\beta_1 = 0.9, \beta_2 = 0.999, \epsilon = 10^{-8}$ with $amsgrad$ set to False. The second optimizer is SGD Loshchilov & Hutter (2017) with momentum=0.9, dampening=0, with Nesterov acceleration. In each experiment, we randomly split the training dataset into 10% validation and 90% training and monitor the validation loss after each epoch. If validation loss did not improve for lr_patience epochs, we reduce the learning rate by multiplying it by lr_factor. We start with an initial learning rate of lr_initial. The training is terminated only if the validation loss did not improve for loss_patience number of epochs or if the training reached 100 epochs. We report the results of the model with the best validation loss.

| Hyper-parameter | LeNet | AlexNet |
| --- | --- | --- |
| optimizer | Adam | SGD |
| minibatch_size | 1000 | 128 |
| lr_initial | 0.0001 | 0.1 |
| lr_patience | 3 | 5 |
| lr_factor | 0.9 | 0.5 |
| loss_patience | 10 | 20 |
| weight_decay | 0 | 0.0005 |

Table 2: Lists the training optimization hyper-parameters.

## B    EXAMPLES ON NOISE LEVELS

Figure 4 provides examples of the different levels of noise on a given digit 8.

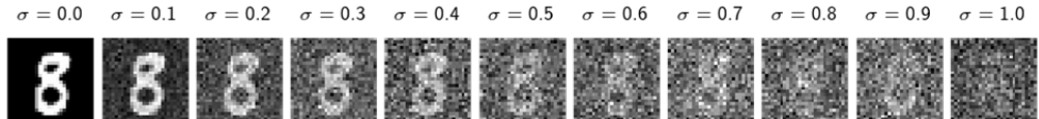

Figure 4: This Figure shows an example of the noise level over varying level of input $\sigma$ on the digit 8. In particular, one can observe that with $\sigma$ large than 0.7 the among of noise is severe even for the human level. Training on such extreme noise levels will deem data augmentation to be difficult.

## C    A COMMENT ON THE ROBUSTNESS METRIC

We measure the robustness against Gaussian noise by averaging over a range of input noise levels, where at each level for each image, we consider it misclassified if the probability of it being mis-classified is greater than a certain threshold. The final robustness is the average over multiple testing $\sigma_x$. This is special case of the more general case in Equation (4). We then report the area under the curve of the robustness with varying testing $\sigma_x$ as shown in Figure 6. The area under this curve thus represents the overall robustness of a given model under several varying input noise standard deviation $\sigma_x$.

## D    OTHER ROBUSTNESS METRICS

We report the robustness of several architectures over several datasets with and without our trained regularizer. We show that our proposed efficient regularizer not only improves the robustness against Gaussin noise attacks but againts several other types of attacks. Table 3 summarizes the types of attacks used for robustness evaluation.

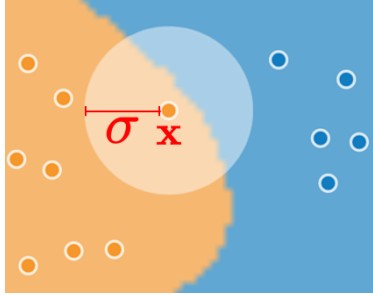

Figure 5: The robustness is a function of the ratio of the orange area to the blue area in the white circle.

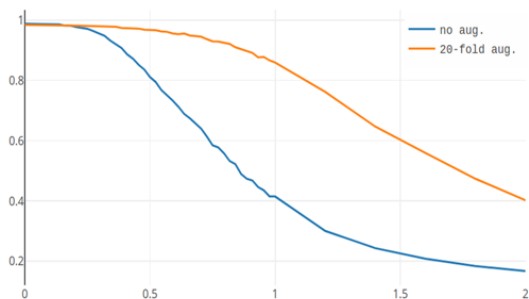

Figure 6: The robustness is thus measured as the area under the curve of testing accuracy versus input noise level (standard deviation).

(a) CIFAR 10 (acc = 68.91%, $\tilde{N} = 11$)         (b) CIFAR 100 (acc = 38.10%, $\tilde{N} = 6$)

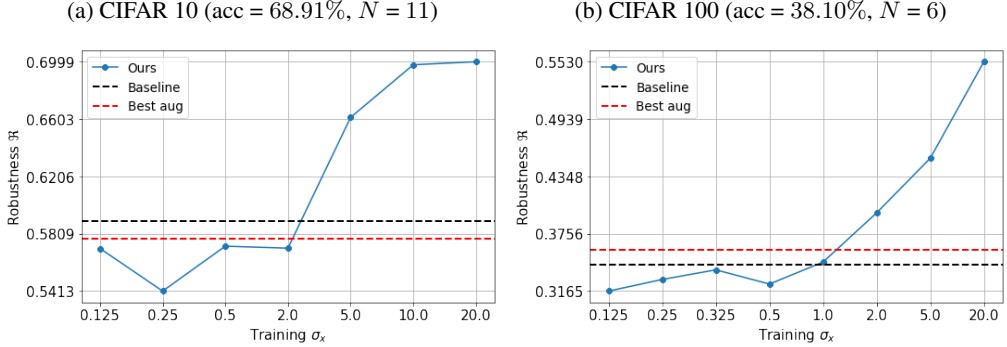

Figure 7: **Fair robustness comparison of AlexNet with data augmentation and our regularizer**. The reported models trained with our regularizer on CIFAR10 and CIFAR100 on all training $\sigma_x$ are within $1.68\%$ and $4.83\%$ of the baseline accuracy, respectively. The models trained with the proposed regularizer achieve better robustness than 11-fold and 6-fold noisy data augmentation on CIFAR10 and CIFAR100, respectively.

| Attack Abbreviation | Attack Name |
|:---:|:---:|
| PGD | Projected Gradient Descent |
| LBF | LBFGS Attack |
| GSM | FGSM |
| AGA | Additive Gaussian Noise Attack |
| AUA | Additive Uniform Noise Attack |
| DF2 | DeepFool $\ell_2$ Attack |

Table 3: The table lists all the attacks performed.

| | $\sigma$ | PGD | LBF | GSM | AGA | AUA | DF2 | GNR | ACC |
|---|---|---|---|---|---|---|---|---|---|
| LeNet on MNIST | 0 | $1.02 \times 10^{-03}$ | $5.50 \times 10^{-04}$ | $2.24 \times 10^{-02}$ | $2.68 \times 10^{-02}$ | $2.49 \times 10^{-02}$ | $5.91 \times 10^{-04}$ | 77.45 | 98.50 |
| | 0.125 | $3.36 \times 10^{-01}$ | $3.43 \times 10^{-01}$ | $8.19 \times 10^{-01}$ | 5.15 | 5.08 | $2.05 \times 10^{-01}$ | 93.14 | 97.50 |
| | 0.250 | $4.58 \times 10^{-01}$ | $4.31 \times 10^{-01}$ | 1.21 | 7.41 | 6.95 | $2.63 \times 10^{-01}$ | 95.64 | 98.75 |
| | 0.325 | $4.21 \times 10^{-01}$ | $4.51 \times 10^{-01}$ | 1.17 | 8.04 | 8.85 | $2.33 \times 10^{-01}$ | 96.75 | 97.50 |
| | 1.0 | $5.44 \times 10^{-01}$ | $5.22 \times 10^{-01}$ | 1.34 | 8.72 | 9.33 | $2.95 \times 10^{-01}$ | 97.32 | 99.00 |
| LeNet on CIFAR10 | 0 | $8.17 \times 10^{-05}$ | $3.38 \times 10^{-06}$ | $1.67 \times 10^{-03}$ | $1.67 \times 10^{-03}$ | $1.71 \times 10^{-03}$ | $3.22 \times 10^{-06}$ | 24.37 | 65.00 |
| | 0.125 | $2.59 \times 10^{-03}$ | $1.74 \times 10^{-03}$ | $3.60 \times 10^{-03}$ | $6.73 \times 10^{-01}$ | $6.99 \times 10^{-01}$ | $1.64 \times 10^{-03}$ | 40.08 | 71.25 |
| | 0.250 | $3.27 \times 10^{-03}$ | $2.05 \times 10^{-03}$ | $4.42 \times 10^{-03}$ | $7.97 \times 10^{-01}$ | $7.80 \times 10^{-01}$ | $1.98 \times 10^{-03}$ | 37.60 | 68.75 |
| | 0.325 | $2.59 \times 10^{-03}$ | $1.66 \times 10^{-03}$ | $3.25 \times 10^{-03}$ | $6.67 \times 10^{-01}$ | $6.56 \times 10^{-01}$ | $1.64 \times 10^{-03}$ | 38.32 | 70.75 |
| | 1.0 | $3.31 \times 10^{-03}$ | $2.14 \times 10^{-03}$ | $4.42 \times 10^{-03}$ | 1.06 | 1.12 | $2.11 \times 10^{-03}$ | 44.03 | 69.75 |
| LeNet on CIFAR100 | 0 | $1.57 \times 10^{-05}$ | $1.33 \times 10^{-05}$ | $4.60 \times 10^{-05}$ | $4.24 \times 10^{-03}$ | $4.46 \times 10^{-03}$ | $1.15 \times 10^{-05}$ | 27.79 | 37.85 |
| | 0.125 | $2.77 \times 10^{-03}$ | $1.62 \times 10^{-03}$ | $3.25 \times 10^{-03}$ | $6.05 \times 10^{-01}$ | $6.00 \times 10^{-01}$ | $1.33 \times 10^{-03}$ | 28.41 | 35.85 |
| | 0.25 | $2.84 \times 10^{-03}$ | $1.55 \times 10^{-03}$ | $3.25 \times 10^{-03}$ | $6.08 \times 10^{-01}$ | $5.96 \times 10^{-01}$ | $1.26 \times 10^{-03}$ | 27.38 | 38.25 |
| | 0.325 | $2.46 \times 10^{-03}$ | $1.42 \times 10^{-03}$ | $2.91 \times 10^{-03}$ | $4.13 \times 10^{-01}$ | $3.83 \times 10^{-01}$ | $1.18 \times 10^{-03}$ | 23.47 | 36.85 |
| | 1.0 | $2.71 \times 10^{-03}$ | $1.63 \times 10^{-03}$ | $3.25 \times 10^{-03}$ | $5.36 \times 10^{-01}$ | $5.69 \times 10^{-01}$ | $1.35 \times 10^{-03}$ | 30.90 | 35.85 |
| AlexNet on CIFAR10 | 0 | $4.55 \times 10^{-03}$ | $1.02 \times 10^{-05}$ | $9.78 \times 10^{-03}$ | $2.93 \times 10^{-03}$ | $2.77 \times 10^{-03}$ | $9.20 \times 10^{-06}$ | 23.49 | 72.75 |
| | 0.125 | $2.08 \times 10^{-03}$ | $1.36 \times 10^{-03}$ | $3.59 \times 10^{-03}$ | $7.96 \times 10^{-01}$ | $8.55 \times 10^{-01}$ | $1.71 \times 10^{-03}$ | 57.08 | 67.50 |
| | 0.250 | $2.05 \times 10^{-03}$ | $1.28 \times 10^{-03}$ | $3.58 \times 10^{-03}$ | $8.05 \times 10^{-01}$ | $8.41 \times 10^{-01}$ | $1.95 \times 10^{-03}$ | 54.13 | 67.00 |
| | 0.325 | $1.75 \times 10^{-03}$ | $1.16 \times 10^{-03}$ | $2.84 \times 10^{-03}$ | $7.53 \times 10^{-01}$ | $7.93 \times 10^{-01}$ | $1.63 \times 10^{-03}$ | 54.03 | 64.50 |
| | 1.0 | $2.09 \times 10^{-03}$ | $1.31 \times 10^{-03}$ | $3.96 \times 10^{-03}$ | $8.72 \times 10^{-01}$ | $8.57 \times 10^{-01}$ | $2.05 \times 10^{-03}$ | 57.86 | 66.50 |
| AlexNet on CIFAR100 | 0 | $2.30 \times 10^{-05}$ | $2.50 \times 10^{-05}$ | $1.50 \times 10^{-05}$ | $4.33 \times 10^{-03}$ | $9.60 \times 10^{-04}$ | $2.10 \times 10^{-05}$ | 29.69 | 34.75 |
| | 0.12 | $3.64 \times 10^{-04}$ | $2.83 \times 10^{-04}$ | $5.06 \times 10^{-04}$ | $1.44 \times 10^{-01}$ | $1.49 \times 10^{-01}$ | $2.16 \times 10^{-04}$ | 31.65 | 33.50 |
| | 0.250 | $4.37 \times 10^{-04}$ | $3.86 \times 10^{-04}$ | $6.50 \times 10^{-04}$ | $1.73 \times 10^{-01}$ | $1.63 \times 10^{-01}$ | $2.47 \times 10^{-04}$ | 32.85 | 32.25 |
| | 0.325 | $5.37 \times 10^{-04}$ | $4.04 \times 10^{-04}$ | $7.29 \times 10^{-04}$ | $1.87 \times 10^{-01}$ | $2.05 \times 10^{-01}$ | $3.18 \times 10^{-04}$ | 33.84 | 34.25 |
| | 1.0 | $4.92 \times 10^{-04}$ | $3.26 \times 10^{-04}$ | $6.50 \times 10^{-04}$ | $1.99 \times 10^{-01}$ | $1.85 \times 10^{-01}$ | $2.85 \times 10^{-04}$ | 34.65 | 35.50 |

Table 4: **Gaussian robustness improves overall robustness.** We report the robustness metrics corresponding to various attacks (PGD, LBFGS, FGSM, AGA, AUA and DF2), our proposed GNR metric, and the test accuracy ACC for LeNet and AlexNet networks trained on MNIST, CIFAR10 and CIFAR100 using our proposed regularizer with noise variance $\sigma$ in training. Note that $\sigma = 0$ corresponds to baseline models trained without our regularizer. We observe that training networks with our proposed regularizer (designed for additive Gaussian attacks) not only improves the robustness against Gaussian attacks but also against 6 other types of attacks.

