# OpenReview forum: "Analytical Moment Regularizer for Training Robust Networks"
_ICLR.cc/2020/Conference — Reject_

### Official Review · AnonReviewer2 · 2019-10-15
**Official Blind Review #2**

**Rating:** 1

**Review:**

This paper proposes a regularizer to encourage the robustness to the random contamination for training deep neural networks. The idea is straightforward and intuitive, but not that exciting. The experiments show some improvement. However, I have a few serious concerns:

(1) Why do we care about the random noise, especially Gaussian noise? There has been a large amount of literature on training robust network, but they are also for the robustness to adversarial examples. The Gaussian noise is too simple and easy to defend. We can even apply a simple denoiser to preprocess the data, which does not even involve training a sophisticated neural network.

(2) The proposed moment regularizer is very delicate. I do not think it can generalize to other noises or contaminations. This is because for other noises, the moment approximation can be fairly loose.

(3) The Alexnet was proposed in 2011. Consider that it is already late 2019, the authors INDEED need to do experiments using more advanced and recent models, e.g., ResNet34/50 or even powerful ones, e.g, ResNeXt, DenseNet, Wide ResNet.

**Experience Assessment:**

I have published one or two papers in this area.

**Review Assessment: Checking Correctness Of Derivations And Theory:**

I assessed the sensibility of the derivations and theory.

**Review Assessment: Checking Correctness Of Experiments:**

I assessed the sensibility of the experiments.

**Review Assessment: Thoroughness In Paper Reading:**

I read the paper thoroughly.

---

> ### Author Response · Authors · 2019-11-15
> **Thank you for taking the time to review our work**
>
> Gaussian data augmentation is useful in many applications. For example, see the cited work by AnonReviewer3. We have also shown how does it improve the robustness against other attacks on LeNet and AlexNet. Due to time constraints, we didn't test it on other architectures.

---

### Official Review · AnonReviewer3 · 2019-10-23
**Official Blind Review #3**

**Rating:** 3

**Review:**

I am not fully convinced by the robustness result in Figure 3. In MNIST, the proposed method is worse than data augmentation. In CIFAR-10, the proposed method does perform better, however, \tilde{N} for data augmentation is chosen as 2, which is too small in my opinion. The data augmentation's robustness is similar to the baseline. I don't know if there are any issues in the training but data augmentation with \tilde{N}=2 is expected to be ineffective to improve robustness. For CIFAR-100, the improvement is marginal and can only be achieved when \sigma is large.

Is the GNR score in Table 1 calculated on all examples or only on correctly classified examples? If it is calculated on all examples, I think it would be better to also report the result of the correctly classified examples. Because at the end we care how the models can correctly and robustly make the classification.

I think it would be better if the authors can have some discussion about the potential connections between the proposed method and the method in [1]. [1] proved that if a model can classify well under Gaussian noise, it is possible to turn it into a classifier that is certifiably robust (I don't think the proposed method is certifiable) to adversarial perturbations in l_2 ball. The training method in [1] is Gaussian data augmentation. An experimental comparison between the proposed method and the method in [1] is necessary. I am not sure how faster the proposed method can be.

[1] Cohen, Jeremy M., Elan Rosenfeld, and J. Zico Kolter. "Certified adversarial robustness via randomized smoothing." ICML 2019

---------------------------
update after rebuttal:
I appreciate the authors' feedback. However, I am not convinced that the proposed method is highly effective (Table 1 and Figure 3) and I still think the contribution is kind of marginal (especially given [1]). So it is hard to recommend acceptance.



**Experience Assessment:**

I have published one or two papers in this area.

**Review Assessment: Checking Correctness Of Derivations And Theory:**

I assessed the sensibility of the derivations and theory.

**Review Assessment: Checking Correctness Of Experiments:**

I assessed the sensibility of the experiments.

**Review Assessment: Thoroughness In Paper Reading:**

I read the paper at least twice and used my best judgement in assessing the paper.

---

> ### Author Response · Authors · 2019-11-15
> **Thank you for taking the time to review our work**
>
>
> - reading Figure 3
>     We are comparing our regularizer, across multiple **training** $\sigma$, against the best-case of data augmentation; namely, the experiment with the highest robustness that was able to maintain the same accuracy as the baseline model, within a specified tolerance. In MNIST, it would be the point on the purple line ($\tilde{N} = 21$) at $\sigma = 0.325$ in Figure 3(a). It is clearly an outlier since we conjecture that this amount of augmentation is enough for MNIST, a relatively low-dimensional data, with this noise level. In CIFAR10, augmentation hurts the accuracy dramatically, and the model that maintained the closest accuracy to the baseline was with $\tilde{N} = 21$. In CIFAR100, the improvement of our regularizer is significant relative to the improvement of the best data augmentation.
> - robustness score
>     We calculate the GNR score on all examples but with the prediction of the model for the clean input image instead of the ground-truth prediction.
> - connection to [1]
>     Thank you for your reference. Indeed, [1] is very relevant to our setup. To perform $\ell_2$ certification, it is necessary to train a base classifier that can classify Gaussianaly corrupted samples correctly. The authors were restricted to performing data augmentation, and as pointed out by authors, this is a suboptimal choice due to reasons that sampling from $\mathcal{N}(\mathbf{0},\sigma^2 \mathbf{I})$ results into samples with an average energy that scales with the dimension of the input. This results into severe difficulties in training. Therefore, our approach with the analytical moment regularizer provides a powerful alternative towards building such a classifier surpassing any need for data augmentation. This is an exciting line of work that we will investigate in a future direction. Regarding speed, please refer to our response to AnonReviewer1.

---

### Official Review · AnonReviewer1 · 2019-10-27
**Official Blind Review #1**

**Rating:** 3

**Review:**

This paper proposes a regularization method for achieving robustness to noisy inputs, with relatively less computation compared to standard data augmentation approaches. Specifically, the authors analyze the analytic expression of the loss on the noisy inputs, and using Jensen’s inequality, propose to minimize a surrogate loss over the expectation of noisy inputs. To minimize the loss over the expectation, the authors impose a regularization over the first moment of the network weights. The authors validate the model with the proposed regularization technique for its robustness against Gaussian attack and other types of attacks, whose results show that the model is robust.

Pros
- The general idea of the regularization that replaces the generation of noisy samples and optimization over it is conceptually appealing and seems practically useful.
- The derivation of the moment-based regularization makes sense.
- The proposed regularizer seems to be effective to a certain degree, on the sets of experiments done by the authors.

Cons
- Experimental validation seems highly inadequate due to lack of baselines. Thus it is difficult to assess the degree of robustness the proposed model achieves. The authors should perform extensive evaluation against state-of-the-art techniques against multiple types of attacks, in order to demonstrate the effectiveness of the proposed method.
- While the authors emphasize the computational efficiency of the method, the authors do not report computational cost or actual runtime.
- The types of non-Gaussian attacks should be better described. Which ones use L-infinity attacks and which use L2 attacks?
- Figure 3 doesn’t seem like a very favorable result to the proposed model, since we are generally more concerned with adversarial examples generated with small perturbations, as large perturbations may change the input semantics.

In sum, while I like the overall idea and find the work novel and potentially practical, it is difficult to properly evaluate the work due to lack of comparison against state-of-the-art data augmentation methods for achieving robustness. Therefore I temporarily give this paper a weak reject, but may change the rating with more experimental results provided in the rebuttal.

**Experience Assessment:**

I have published one or two papers in this area.

**Review Assessment: Checking Correctness Of Derivations And Theory:**

I assessed the sensibility of the derivations and theory.

**Review Assessment: Checking Correctness Of Experiments:**

I carefully checked the experiments.

**Review Assessment: Thoroughness In Paper Reading:**

I read the paper at least twice and used my best judgement in assessing the paper.

---

> ### Author Response · Authors · 2019-11-15
> **Thank you for taking the time to review our work**
>
>
> - comparisons
>     Due to time constraints, we couldn't compare with other baselines during the rebuttal period. Nevertheless, the main motivation behind this work is to demonstrate the effectiveness of our regularizer as an efficient replacement for data augmentation that works even with a high noise regime on high dimensional data. The reported robustness improvement is one way to show this.
> - computational cost
>     We averaged the time it takes to do 10 training epochs on AlexNet with and without our regularizer. Our method has an overhead that is equivalent to doing data augmentation with $\tilde{N} \approx 1.83$. Of course, this number is inversely proportional to the depth of the network, demonstrating the usefulness of this method.
> - attacks norms
>     Every robustness metric can be investigated independently but we should clarify the reported units. Thank you for pointing this out.
> - favorable result
>     The x-axis in Figure 3 is for the training noise level. Whereas, the reported robustness is averaged over 30 evenly sampled noise levels in [0, 0.5]. We show in Appendix B examples of different noise levels applied to an MNIST image. It is indeed favorable, considering how we can achieve a sweet tradeoff between accuracy and robustness.

---

### Author Response · Authors · 2019-09-30
**Code**

It is far unlikely unless purposely looked for, the affiliation of the authors can be exposed if the meta data of the provided code is thoroughly investigated.

To this end, we disable the shared link previously provided and provide the new link here

https://drive.google.com/file/d/15ogVaejLjx53T2aMlHoa1Q-b52jdSOwY/view

---

### Decision · Program_Chairs · 2019-12-19

**Decision:**

Reject

**Comment:**

This paper received two weak and one strong reject from the reviewers.  The major issues cited were 1) a lack of strong enough baselines or empirical results, 2) Novelty with respect to "Certified adversarial robustness via randomized smoothing" and 3) a limitation to Gaussian noise perturbations.  Unfortunately, as a result the reviewers agreed that this work was not ready for acceptance.  Adding stronger empirical results and a careful treatment of related work would make this a much stronger paper for a future submission.